# Combination of pre-adapted bacteriophage therapy and antibiotics for treatment of fracture-related infection due to pandrug-resistant *Klebsiella pneumoniae*

Anaïs Eskenazi [1,9 ✉], Cédric Lood [2,3], Julia Wubbolts[4], Maya Hites[1], Nana Balarjishvili[5], Lika Leshkasheli[5], Lia Askilashvili[5], Leila Kvachadze[5], Vera van Noort [3,6], Jeroen Wagemans [2], Marc Jayankura[7], Nina Chanishvili [5], Mark de Boer[4], Peter Nibbering[4], Mzia Kutateladze[5], Rob Lavigne [2], Maya Merabishvili[8] & Jean-Paul Pirnay [8]

A 30-year-old bombing victim with a fracture-related pandrug-resistant *Klebsiella pneumoniae* infection after long-term (>700 days) antibiotic therapy is treated with a pre-adapted bacteriophage along with meropenem and colistin, followed by ceftazidime/avibactam. This salvage therapy results in objective clinical, microbiological and radiological improvement of the patient's wounds and overall condition. In support, the bacteriophage and antibiotic combination is highly effective against the patient's *K. pneumoniae* strain in vitro, in 7-day mature biofilms and in suspensions.

[1] Clinic of Infectious Diseases, CUB-Erasme Hospital, Brussels, Belgium. [2] Department of Biosystems, Laboratory of Gene Technology, KU Leuven, Leuven, Belgium. [3] Department of microbial and molecular systems, Computational systems biology, KU Leuven, Leuven, Belgium. [4] Department of Infectious diseases, Leiden University Medical Centre, Leiden, The Netherlands. [5] Eliava Institute of Bacteriophages, Microbiology and Virology, Tbilisi, Georgia. [6] Institute of Biology, Leiden University, Leiden, The Netherlands. [7] Department of Orthopedic Surgery, CUB-Erasme Hospital, Brussels, Belgium. [8] Laboratory for molecular and cellular technology, Queen Astrid Military Hospital, Brussels, Belgium. [9] Present address: Centre Hospitalier de Cayenne, Cayenne, French Guiana. ✉email: anaiseskenazi@me.com

The increasing number of cases of fracture-related infection (FRI) due to multidrug-resistant (MDR) bacteria limits treatment options and surgical re-interventions are often necessary[1]. Among the MDR nosocomial pathogens, *Klebsiella pneumoniae* is a major concern because of its ability to form biofilms, to avoid the immune system, and to acquire genes coding for rapidly evolving enzymes, such as extended-spectrum β-lactamases (ESBLs) and carbapenemases[2]. Biofilms are structures elaborated by bacterial communities attached to the surface of implants or avascular tissue fragments. Within biofilms, persister cells form a subpopulation of metabolically dormant cells that play a major role in the capacity of biofilms to survive and recover from antibiotic treatment[3]. The treatment of FRI caused by MDR bacteria is very challenging, as it increasingly requires long-term administration of high-doses of broad-spectrum antibiotics, often associated with reversible and irreversible adverse events and low success rates. Phage therapy, the use of (bacterio)phages (viruses that infect bacteria) to treat bacterial infections, is emerging as an additional tool in the fight against MDR bacteria[4]. Usually, phages only infect a subset of strains belonging to a single bacterial species. That is why personalized phage therapy approaches use one or more phages selected from a preprepared bank, based on their lytic activity against the bacterial strains isolated from the patient's infection. When time and resources permit, the selected phages can be pre-adapted or trained, i.e., phage mutants with increased infectivity and reduced capacity to provoke bacterial resistance are selected in vitro[5]. The last decade has seen a surge in phage therapy research, including the use of phages to treat orthopedic-device-related infections[6].

On March 22, 2016, a 30-year-old woman suffered a polytrauma during the suicide bombing at the Brussels Airport. Upon admission to the intensive care unit of the Erasme Hospital, she presented a cardiac arrest secondary to hemorrhagic shock caused by heavy bleeding from explosion wounds on her left flank and thigh. After an aggressive multidisciplinary intervention, including reanimation, partial amputation of the iliac bone, and external fixation of a broken femur, the patient was stabilized (Fig. 1).

However, on day 4, the patient progressed to a septic shock due to a surgical wound infection of the left thigh, despite antibiotic treatment with amoxicillin/clavulanate upon admission, followed by piperacillin/tazobactam. Bacterial culture of surgical biopsies showed a polymicrobial flora, consisting of *Enterococcus faecium*, *Pseudomonas aeruginosa*, *Enterobacter cloacae*, and *K. pneumoniae*. This was the starting point of a long-term, high-dose, broad-spectrum poly-antibiotic therapy guided by therapeutic drug monitoring (Fig. 1). To control the infection, a grand dorsal muscle autograft was used to close the major blast wound on the left leg, 14 days after admission. However, the graft immediately necrotized superficially, requiring further debridement. The high-dose and prolonged antibiotic treatment led to adverse events, resulting in the premature stopping of certain antibiotics: febrile neutropenia due to meropenem, aggravation of the patient's post-traumatic deafness due to amikacin, and renal tubulopathy due to colistin. To complicate matters even further, the patient was diagnosed with invasive mucormycosis attributed to immune dysfunction due to severe critical illness, 25 days after admission. Extensive necrosis was observed and complete gastrectomy and splenectomy were performed[7]. A 6-week course of high-dose antifungals (Fig. 1) and an experimental treatment with nivolumab, a cell death protein 1 (PD-1) antagonist, and IFNγ were initiated[8]. After more than 4 months of intensive antibiotic therapy, treatment was suspended and the patient was monitored closely. A skin autograft was applied on the left thigh wound in July 2016. However, shortly after grafting, a sinus tract with purulent discharge appeared in the middle of the skin graft. Pus

also discharged from the orifices of the external-fixator pins. FRI facilitated by the remaining small bone fragments, the retained metal shrapnel from the bomb, and the external fixator was suspected. The external fixator was replaced and the wound was debrided 170 days post-injury. Surgical biopsies (from the femur) showed the presence of 2 *K. pneumoniae* strains (Fig. 1 and Table 1), one of them (Kp040762) exhibiting an extensively drug-resistant (XDR) phenotype; i.e., it was non-susceptible to at least 1 agent in all but 2 antimicrobial categories[9]. Two other microorganisms were found in the biopsies: multi-susceptible *S. aureus* and *Mycobacterium xenopi*. An adapted antibiotic regimen was started on day 175 (Fig. 1).

Long-term antibiotic treatment failed to cure the FRI, translating into delayed wound healing and complete absence of femur consolidation. Confronted with this therapeutic failure, the clinicians decided to turn to phage therapy. The 2 day-170 *K. pneumoniae* isolates (Kp040741 and Kp040762) were sent to the Eliava Institute in Tbilisi (Georgia) to select and pre-adapt a therapeutic phage. Phage vB_KpnM_M1 (M1) was shown to exhibit the highest activity against the patient's *K. pneumoniae* isolates. It was isolated from a sewage water sample in Tbilisi, in 2012. Transmission electron microscopy revealed a myovirus morphology (Fig. 2a). Phage M1 showed a broad host range (~65%) against clinical isolates of different species of the genus *Klebsiella* (Fig. 2b). A one-step growth experiment revealed that phage M1 has a productive reproduction cycle with a short adsorption period (91% of phage particles adsorbed in 2 min, 99% in 6 min) (Fig. 2c), causing bacterial cell lysis within 8–10 min after a latent period of 35 min, and an average burst size of 43 phage particles per infected cell (Fig. 2d). Maximal phage activity was observed over the pH range of 5.0–10.0 (Fig. 2e) and the phage was thermostable until 50 °C (Fig. 2f).

Fifteen rounds of co-evolution using Appelmans' method[10] significantly reduced the incidence of bacterial phage resistance. Stable lysis (i.e., without the emergence of phage insensitive mutants) of the patient's *K. pneumoniae* strains was achieved in broth for at least 6 h, at a concentration of $10^3$ plaque-forming units (PFU)/ml and a multiplicity of infection (MOI) of 0.0001. Whole-genome sequencing showed that this myovirus belongs to the *Slopekvirus* genus within the *Tevenvirinae* subfamily, having type species *Klebsiella* phage KP15 as its closest relative (Fig. 3a)[11]. A sequence-based comparison between the pre-adapted and the originally selected variants of phage M1 suggests that a missense mutation (Thr281Arg) in the loop region of the hinge connector of the distal tail fiber protein (Fig. 3b and Supplementary Data 1), which may cause alterations in the phage receptor, is at the basis of the observed improved lytic activity against the patient's day-170 *K. pneumoniae* isolates. However, conclusive proof of this genotype-phenotype association would require engineering the originally selected phage M1 variant to reintroduce this mutation independently, which we have not been able to achieve to date. No known depolymerase genes were detected in the genome of phage M1. Importantly, its genome does not encode proteins associated with toxicity, lysogeny or antibiotic resistance; the absence of these protein-coding genes being required for a phage to be used in phage therapy in Belgium[12].

The ethical committee of Erasme Hospital gave clearance for the use of experimental phage therapy in XDR *K. pneumoniae* FRI in November 2016 and the patient signed an informed consent. However, due to a lack of consensus among the treating physicians, phage therapy was put on hold. On 21 February 2018 (702 days postinjury), facing a therapeutic dead end, the phage preparation was applied at the end of a surgical intervention, which consisted of radical debridement followed by the transplantation of rifampicin-impregnated autologous bone grafts

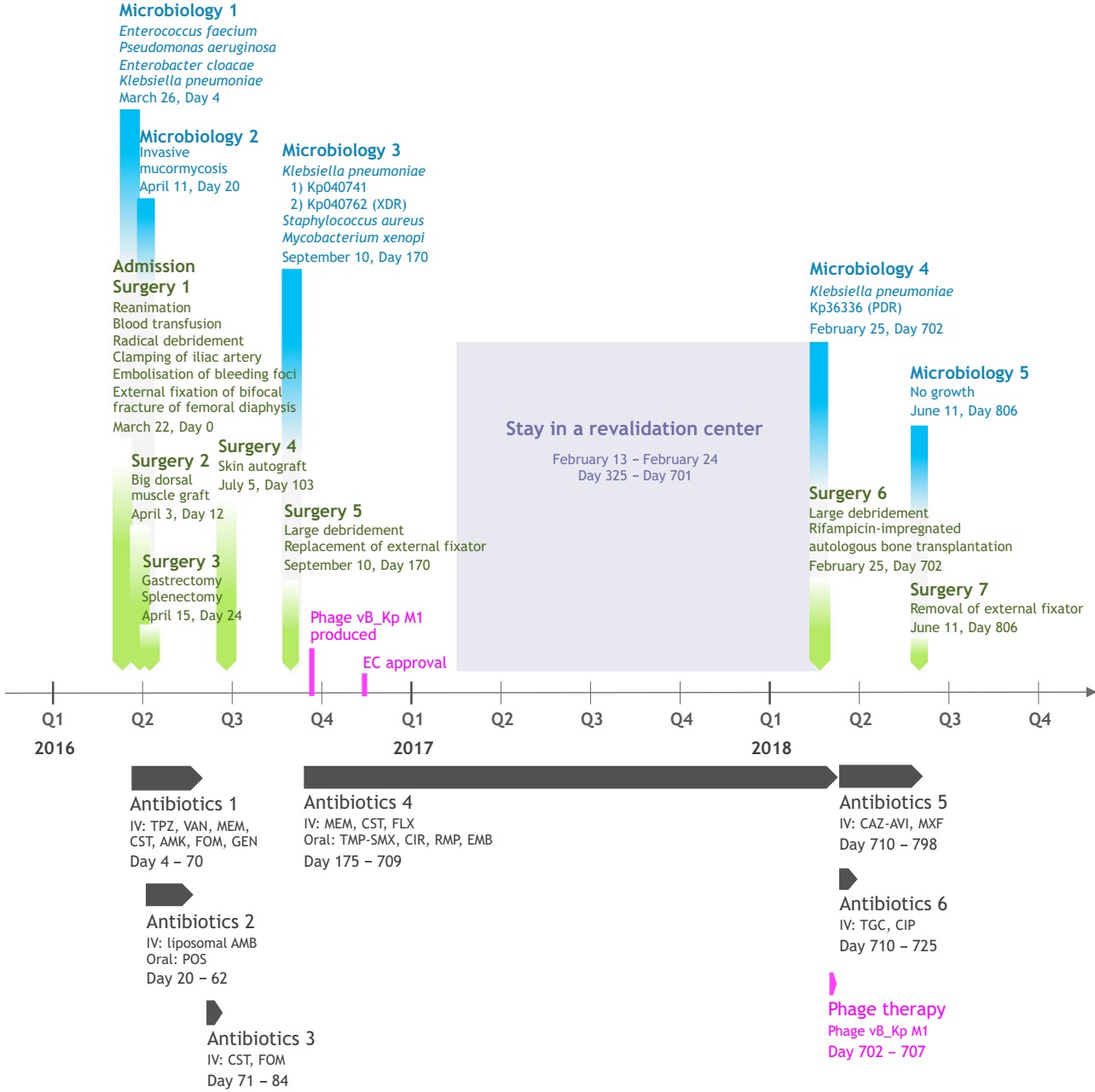

**Fig. 1 Timeline of the most relevant surgical procedures (green), microbiological results (blue), antibiotic therapies (dark grey), and phage therapy (magenta).** Phage therapy related events are indicated in fuchsia. AMB, liposomal amphotericin-B (600 mg, q24h); AMK, amikacin (1800 mg, q24h); CAZ-AVI, ceftazidime/avibactam (2 g/0.5 g, q8h); CIP, ciprofloxacin (400 mg, q8h); CIR, clarithromycin (500 mg, q12h); CST, colistin (up to 20 MIU, q24h); EMB, ethambutol (1200 mg, q24h); FLX, flucloxacillin (2 g, q4h); FOM, fosfomycin (4 g, q4h); GEN, gentamicin (400 mg, q24h); MEM, meropenem (up to 2 g, q4h); POS, posaconazole (200 mg, q8h); TMP-SMX, trimethoprim/sulfamethoxazole (1660 mg, q12h); MXF, moxifloxacin (400 mg, q24h); PDR, pandrug-resistant; RMP, rifampicin (450 mg, q24h); TGC, tigecycline (100 mg, q12h); TPZ, piperacillin/tazobactam (4.5 g, q6h); VAN, vancomycin (30 mg/kg/day); XDR, extensively drug-resistant.

(IABGs) (Fig. 1). In the absence of robust pharmacological data with regard to the use of phages in FRI and without medical consensus for intravenous administration, phages were applied locally (through a catheter left in place) to obtain the highest concentration of phages at the site of infection. A short course of treatment (6 days) was chosen to minimize possible immunogenicity and the ongoing antibiotic therapy was continued (Fig. 1).

Here, we show that salvage therapy consisting of a preadapted phage along with meropenem and colistin, followed by ceftazidime/avibactam results in clinical, microbiological and radiological

improvement of the patient's wounds and overall condition. We provide evidence that the phage and antibiotic combination is highly effective against the patient's *K. pneumoniae* strain in vitro, in suspensions as well as in biofilms.

## Results and discussion

Susceptibility testing of a day-702 *K. pneumoniae* isolate from a biopsy taken during surgery showed a pandrug-resistant (PDR) status (isolate Kp36336 in Table 1); i.e., non-susceptibility to all agents in all antimicrobial categories[9]. Sequencing analysis showed that all *K. pneumoniae* isolates belonged to sequence type

**Table 1 Minimum inhibitory concentrations (MICs, μg/ml) and EUCAST-based categorization of the susceptibility (using the VITEK 2 system) of a selection of the patient's *Klebsiella pneumoniae* isolates to a selection of antibiotics.**

| Antibiotics | *K. pneumoniae* isolates (days after the injury, on March 22, 2016) | | | |
| --- | --- | --- | --- | --- |
| | Kp1 (4) | Kp040741 (170) | Kp040762 (170) | Kp36336 (702) |
| Ampicillin | R** | ≥ 32 (R) | ≥ 32 (R) | ≥ 32 (R) |
| Amoxicillin-clavulanic acid | R** | ≥ 32 (R) | ≥ 32 (R) | ≥ 32 (R) |
| Piperacillin-tazobactam | R** | ≥ 128 (R) | ≥ 128 (R) | ≥ 128 (R) |
| Temocillin | 256 (R) | ≥ 32 (R) | ≥ 32 (R) | ≥ 32 (R) |
| Cefuroxime | R** | ≥ 64 (R) | ≥ 64 (R) | ≥ 64 (R) |
| Ceftazidime | ≥ 64 (R) | 2 (R) | 32 (R) | ≥ 64 (R) |
| Cefotaxime | ≥ 64 (R) | 8 (R) | ≥ 64 (R) | ≥ 64 (R) |
| Cefepime | ≥ 64 (R) | 2 (R) | ≥ 32 (R) | ≥ 32 (R) |
| Meropenem | ≥ 32 (R) | ≥ 16 (R) | ≥ 16 (R) | ≥ 16 (R) |
| Ertapenem | ≥ 32 (R) | ≥ 8 (R) | ≥ 8 (R) | ≥ 8 (R) |
| Gentamicin | < 1 (S) | ≤ 1 (S) | ≥ 16 (R) | ≥ 16 (R) |
| Amikacin | 16 (I) | ≤ 2 (S) | 4 (S) | 16 (I) |
| Ciprofloxacin | 2 (R) | ≤ 0.25 (S) | ≥ 4 (R) | 1 (R) |
| Moxifloxacin | ND | 22 (S)* | 10 (R)* | 18 (R)* |
| Tigecycline | 0.5 (S)* | 20 (I)* | 18 (R)* | 16 (R)* |
| Fosfomycin | S** | 128 (R) | 32 (S) | ≥ 256 (R) |
| Colistin | 0.25 (S) | ≤ 0.5 (S) | ≥ 16 (R) | ≥ 16 (R) |

a No other information than R or S could be recovered from the patient's medical file. I, intermediate (orange); *ND* not determined (white), *R*, resistant (yellow), *S* susceptible (blue).
b Inhibition zone diameters (mm) and EUCAST-based categorization of antibiotic susceptibilities obtained by the disk diffusion method.

ST893, which was associated with outbreaks in Iran in the period 2014–2016[13], and harbored a predicted capsular type K20. Isolates Kp040741 and Kp040762 originated from the same wound sample, but displayed different colony morphologies and antibiotic resistance profiles (Table 1). Analysis of their genomes revealed that both isolates were clonal, but had different plasmid contents, with isolate Kp040741 containing 3 plasmids, compared to the 5 plasmids of isolate Kp040762 (Figs. 4 and 5). One of the plasmids (pSID3), present in all 3 isolates, appeared to be a prophage element similar to bacteriophage SSU5 (Supplementary Data 2), which was previously suggested to be the phylogenetic origin of cryptic plasmid pHCM2 harbored by a *Salmonella* Typhi strain[14]. SNP analysis indicated limited within-host adaptation (Supplementary Data 3). All isolates were shown to possess the pOXA48 plasmid, which contains the class D carbapenem-hydrolyzing β-lactamase OXA-48[15]. Multiple antibiotic resistance genes (ARGs) were observed in the different isolates, often in multiple copies, with PDR isolate Kp36336 containing the most ABR genes (Fig. 5 and Supplementary Data 4). The strain harbored no less than 4 widely distributed ESBL genes (*bla*SHV-145, *bla*OXA-1, *bla*TEM-1, and *bla*CTX-M-15), 5 aminoglycoside modifying enzyme (AME) genes [*aac(6')-Ib*, *aph(3")-Ib*, *aadA1*, *aac(3)-IIe*, and *aph(6)-Id*)], quinolone resistance gene *qnrB1*, and the multidrug resistance *oqxA5* and *oqxB19* pump genes. Trimethroprim/sulfamethoxazole (*dfrA14* and *sul2*), tetracycline (*tetA*), fosfomycin (*fosA6*), and chloramphenicol (*catA1*) resistance genes were also identified. Fortunately, the *K. pneumoniae* phage that had been pre-adapted to target the 2 day-170 isolates was equally active against the day-702 isolate (Fig. 6). Even though the patient showed signs of improvement within 2 days of starting phage therapy, the long-standing antibiotic therapy was replaced by ceftazidime/

avibactam, which had just become available through a compassionate use program, complemented with high-doses of tigecycline and ciprofloxacin (Fig. 1). With a minimal inhibitory concentration (MIC) of 2.0/0.5 μg/ml, the day-702 isolate was found to be susceptible to ceftazidime/avibactam. Ceftazidime/avibactam was administered at the recommended dosage (2 g/0.5 g, q8h), which was predicted to lead to peak plasma concentration of 90.4 μg/ml and 14.6 μg/ml for ceftazidime and avibactam, respectively (according to the FDA-approved labeling information).

Two weeks later, the patient developed grade A acute pancreatitis due to tigecycline. Therefore, the drug was suspended and symptoms resolved. Oral ciprofloxacin was replaced by moxifloxacin (first intravenously, then orally) due to persisting nausea. The adapted antibiotic treatment was administered for a period of 3 months (Fig. 1).

No adverse events associated with the use of phages were observed. Three months after the day-702 surgical intervention and initiation of the combined phage-antibiotic therapy (Fig. 1) the patient's clinical, biological, microbiological and radiological conditions were assessed and compared to the pre-surgery status (Fig. 7a). The skin graft appeared vascularized and viable, the sinus tract was closed and dry, and pus no longer discharged from the pin sites of the external fixator (Fig. 7b), indicating that the FRI was controlled[16]. The patient's general condition had also improved, with a weight gain of 5 kg and a restored muscle function of the left thigh. Biological blood parameters were within normal reference values. A computed tomography (CT) scan of the left femur showed partial consolidation of the fractures (Fig. 7c). The patient's immune system began to synthesize neutralizing antibodies sometime between days 8 and 18 post phage application (Fig. 7d). It is thus highly unlikely that phage-

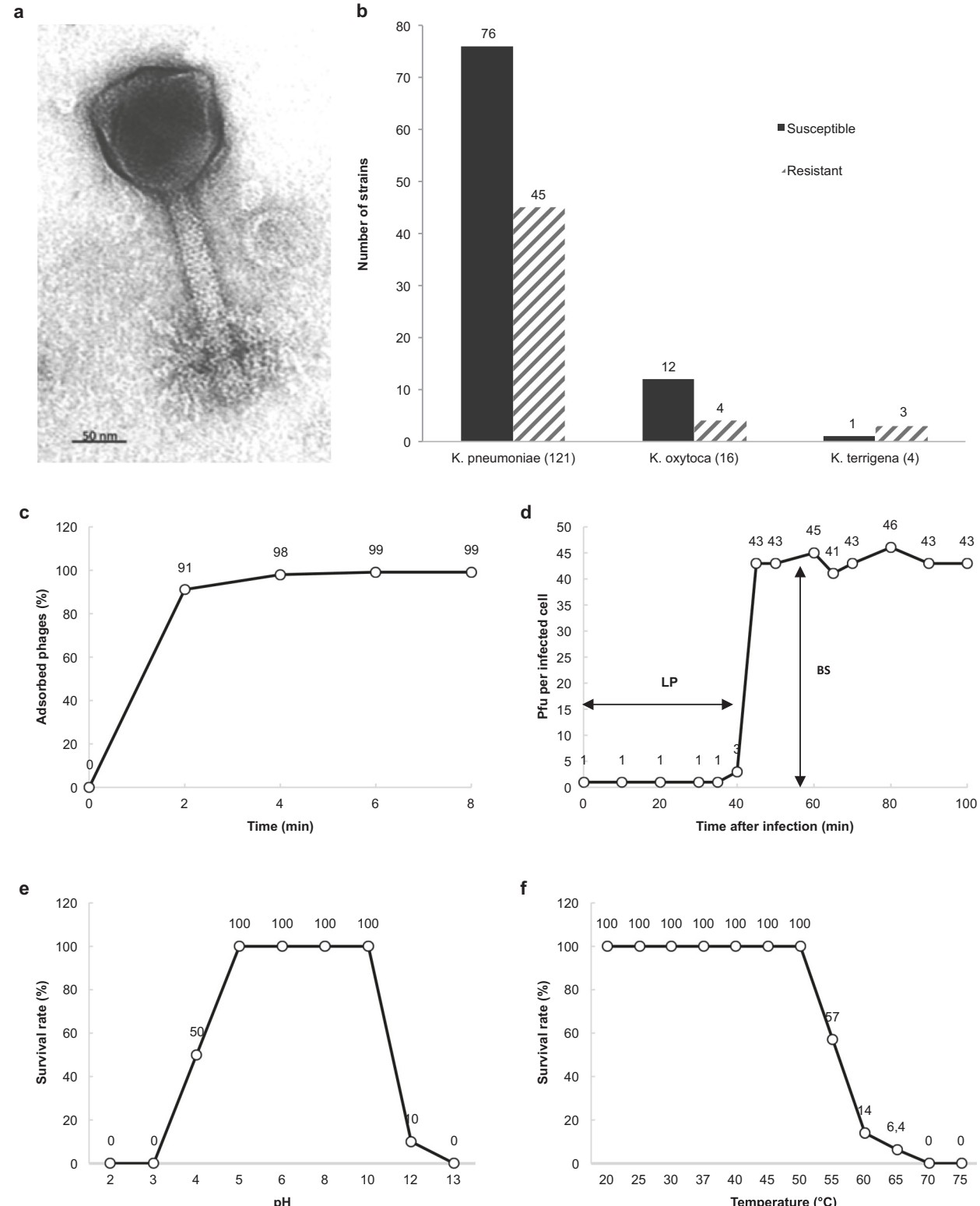

**Fig. 2 Characteristics of phage M1.** Transmission electron micrograph showing an icosahedral head (120 × 120 nm) and a contractile tail (150 × 22 nm) (**a**). Lytic activity of phage M1 against 141 *Klebsiella* spp. clinical isolates obtained from Georgia (*n* = 88), France (22), Switzerland (21), Singapore (6), and China (4) (**b**). Biological characteristics of phage M1 as determined in propagation strain 1a. Adsorption curve (**c**), single-step growth curve (**d**), pH stability (**e**), and temperature stability (**f**). BS, burst size; LP, latent period; PFU, plaque-forming unit. Source data are provided as a Source Data file.

neutralizing antibodies hampered the course of phage therapy. Of interest, on February 23, 2021, 3 years after phage application, the patient's serum no longer contained phage-neutralizing antibodies.

In the absence of clinical signs of FRI, it was decided to discontinue antibiotic treatment 3 months post-surgery and 798 days post-injury (Fig. 1). A week later, the external fixator was removed. No biopsies were performed, to not disturb the

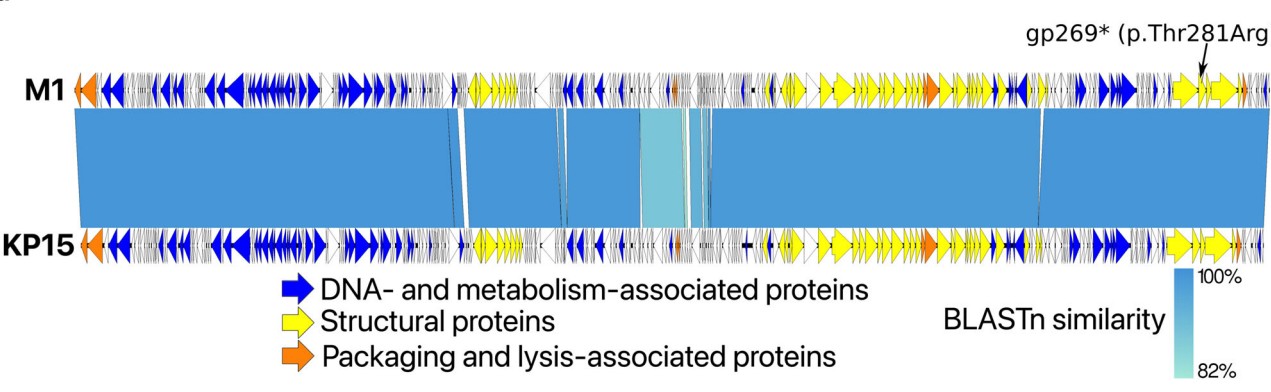

**Fig. 3 Relevant genomic and proreomic characteristics of phage M1.** Genome representation of phage M1 and comparison, using a BLASTn analysis, with *Slopekvirus* type species KP15 (**a**). Each white or colored arrow represents a predicted open reading frame. In orange, genes encoding packaging and lysis-associated proteins are displayed, in yellow structural proteins and in blue DNA- and metabolism-associated proteins (adapted from EasyFig 2.2.2). The single missense mutation found in the preadapted phage M1 isolate used for therapy is indicated (gp269). AlphaFold2 model of the hinge connector of the distal tail fiber (**b**). The mutation of a Threonine to an Arginine is indicated (position 94). Alpha helices are indicated in red and beta sheets in yellow. Green stretches exhibit no specific structures. The model quality scores are presented to the right of the model. lDDT local distance difference test.

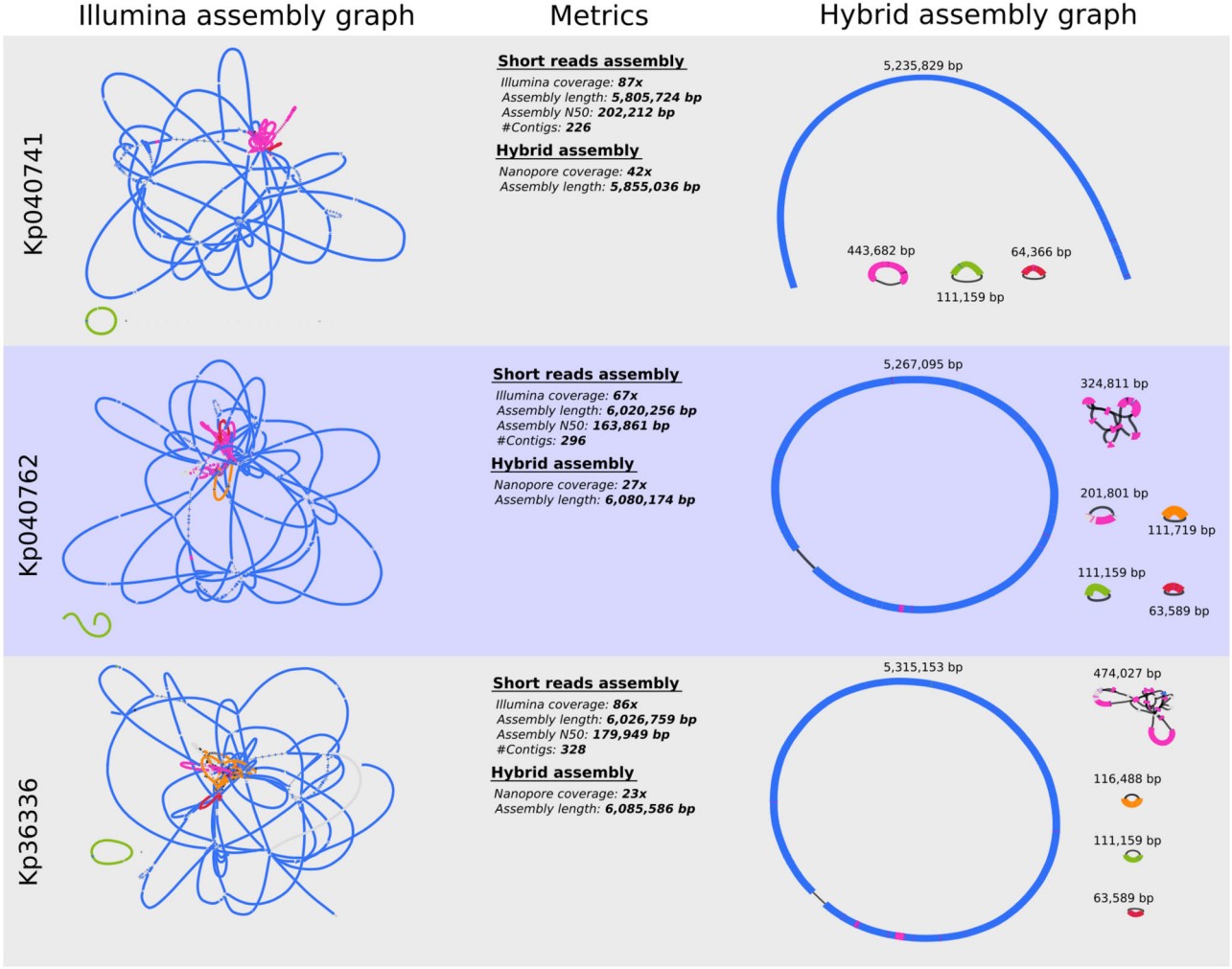

**Fig. 4 Bandage visualization of the short-reads (Illumina) and hybrid (Illumina + Nanopore) assembly graphs of *Klebsiella pneumoniae* isolates Kp040741, Kp040762, and Kp36336.** The different replicons can be distinguished in the hybrid assembly graphs and include the chromosome and up to five plasmids (isolate Kp040762).

fragile consolidation, but 3 bone fragments that detached while removing the external fixator were collected for bacterial culture. These cultures, as well as several subsequent wound cultures, showed no growth. There was no sign of persistent or recurrent *K. pneumoniae* infection after discontinuation of antibiotics (Fig. 1). At the time of writing, 3 years after phage-antibiotic combination treatment, the patient has regained ambulation and mobility, usually with the aid of crutches, and participates in sporting events (e.g., cycling)[17], and there are no signs of recurrent *K. pneumoniae* infection.

The present case report illustrates the complexity of management of FRI caused by MDR bacteria and the urgent need for new antimicrobial agents. Pre-adapted phage-antibiotic combination therapy ultimately led to a positive clinical outcome of a long-standing PDR *K. pneumoniae* FRI. Concordantly, in vitro data indicates that the phage-antibiotic combination was more effective in reducing bacterial counts for *K. pneumoniae* in mature biofilms than antibiotics or phage alone (Fig. 8). We observed that ceftazidime/avibactam dose- and time-dependently reduced bacterial counts for *K. pneumoniae* in mature biofilms, but without complete eradication even at the highest concentration that can be achieved in the systemic circulation (Cmax, 90/ 22.5 µg/ml). High doses of the phage did not eradicate the bacteria in such mature biofilms either. However, combinations of high doses of pre-adapted phage M1 and moderate

concentrations of ceftazidime/avibactam were significantly more effective than the antibiotic alone.

As planktonic bacteria also play a role in infection, we investigated whether a synergistic activity of preadapted phage M1 and administered antibiotics could also be observed in suspension (in vitro). Results revealed that combinations of preadapted phage M1 and ceftazidime/avibactam (Fig. 9a–c) and meropenem (Fig. 9d), but not colistin (Fig. 9e), were considerably more effective against PDR *K. pneumoniae* isolate Kp36336 than phage M1 or these antibiotics alone. Considering the clinical data and the in vitro phage-antibiotic synergy data, there are solid indications that the combination of phage M1 and the antibiotics meropenem and ceftazidime/avibactam finally led to the clinical resolution of infection in this patient.

The current state of the art with regard to infections with carbapenem-resistant *Enterobacteriaceae* advocates the use of combination therapies to prevent or delay the emergence of resistance during antimicrobial therapy[18]. In a mouse model of diabetic foot ulcers infected with methicillin-resistant *S. aureus*, the combination of lytic phages and linezolid was more effective in controlling the infection than phages or linezolid alone[19]. Phages were also shown to infect persister cells in biofilms and turn them into metabolically active cells prior to killing them[20]. Recently, administration of phages was associated with a satisfactory clinical outcome in a case of intractable biofilm-associated

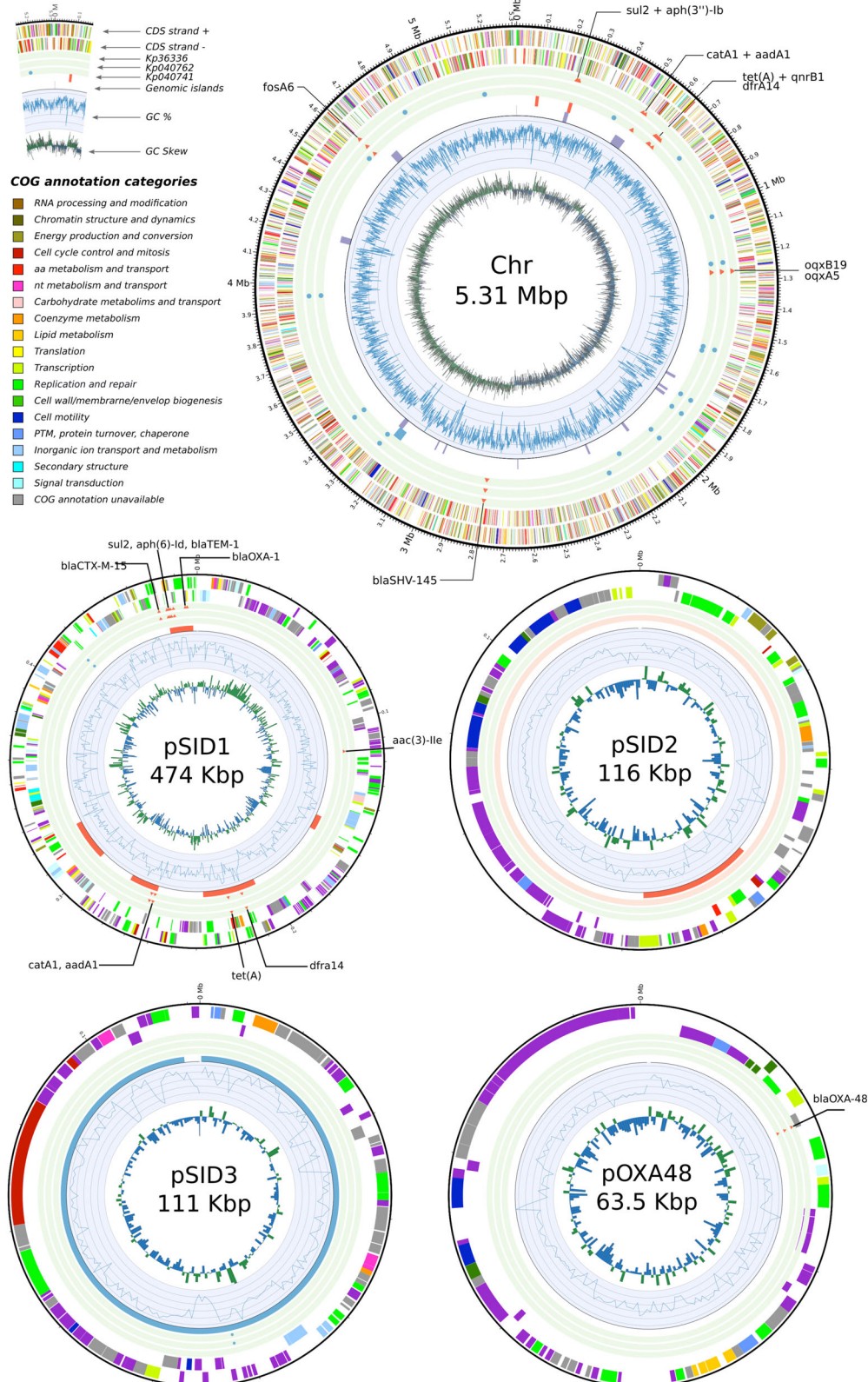

**Fig. 5 Visualization of the functional content of the pandrug-resistant *Klebsiella pneumoniae* isolate Kp36336 among the different replicons.** The antibiotic resistance genes (ARGs) and single nucleotide polymorphisms (SNPs), as well as the prophage content and genomic islands, found in the three distinct *K. pneumoniae* isolates were added (Supplementary Data 2–4).

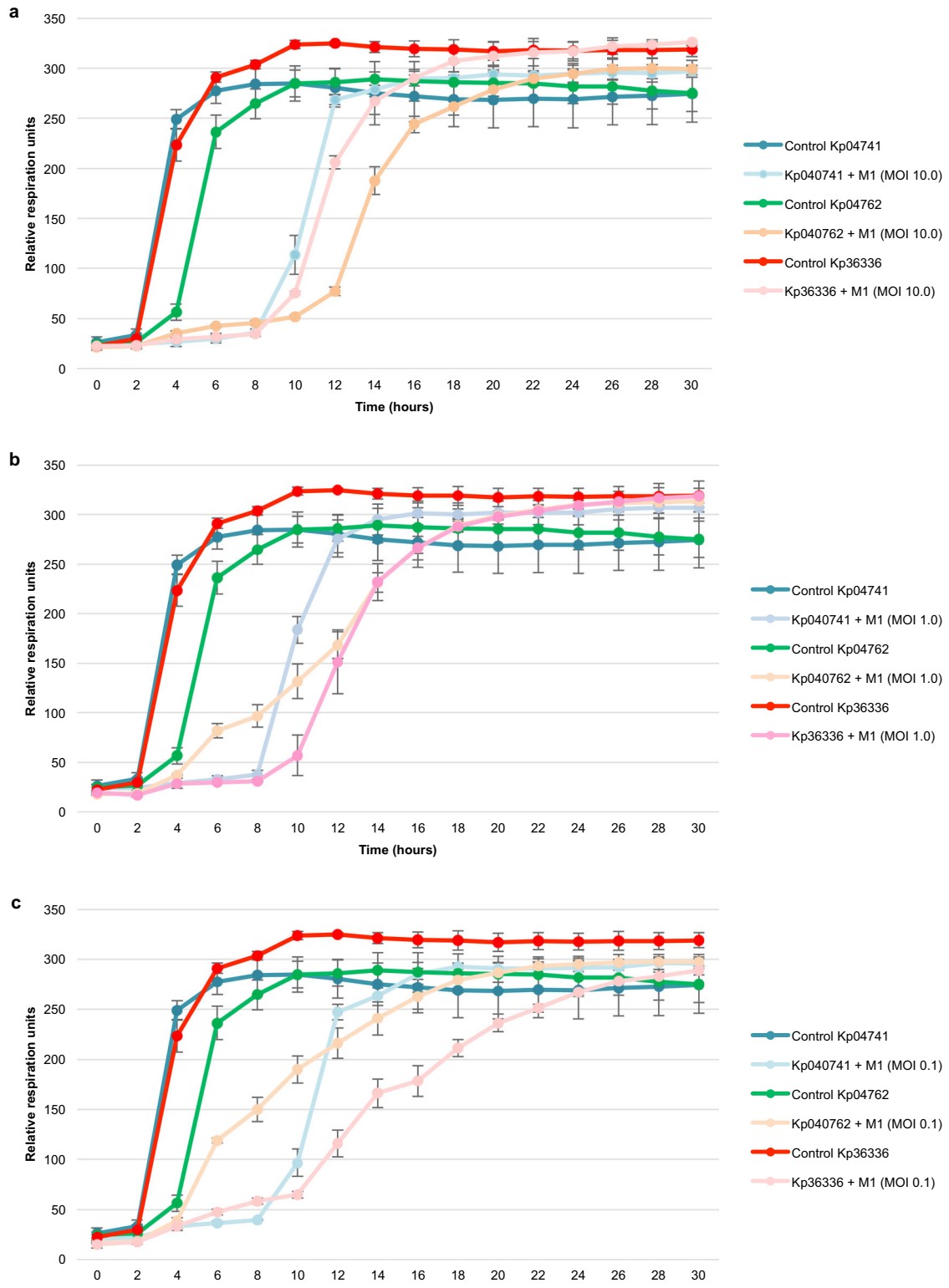

**Fig. 6 Pre-adapted phage M1 activity.** The activity of pre-adapted phage M1 against *Klebsiella pneumoniae* isolated on day 170 (isolates Kp04741and Kp04762) and day 702 (isolate Kp36336) postinjury were determined. Phage activity was measured at different multiplicities of infection (MOI): MOI 10.0 (**a**), MOI 1.0 (**b**), and MOI 0.1 (**c**). Controls consisted of the *K. pneumoniae* isolates without phages. The curves represent bacterial proliferation (cellular respiration). Results are presented as mean values of biological replicates ($n = 3$) with error bars representing the standard deviations of the means. Source data are provided as a Source Data file.

**a**

| Condition | Before surgery | Three months post-surgery |
|---|---|---|
| *Clinical* | | |
| Skin graft appearance | Red, thick | Pink |
| Sinus tract appearance | Discharge of pus | Dry, closed |
| *Biological [normal levels]* | | |
| CRP [< 10 mg/ml] | 2.8 | 8 |
| ESR [2-20 mm/H] | 36 | No data |
| WBC [4.2-11.4x10$^3$/mm$^3$] | 8.1 | 10.3 |
| *Microbiological* | | |
| Bacterial culture | 5 of 11 surgical bone biopsies grew PDR *K. pneumoniae* | From 15 days post-surgery on, none of the samples showed bacterial growth |
| *Radiological* | | |
| CT scan left femur fractures | No consolidation of the bifocal fracture of the upper diaphysis | Multiple foci of consolidation |

**b**

Before treatment    Post treatment

**c**

Admission    June 2017    February 2018

June 2018    June 2019

**d**

**Fig. 7 Comparison of the patient's condition before and after combined phage-antibiotic treatment.** Relevant clinical, biological, microbiological, and radiological parameters before (within 3 days before, or – for microbiological parameters – during surgery) and after treatment (**a**). CRP, C reactive protein; CT, computed tomography; ESR, erythrocyte sedimentation rate; PDR pandrug-resistant, WBC white blood cells. Antero-external view of the patient's left hip and thigh 18 months before and 3 months after treatment (**b**). Successive computed-tomography scanners of the left femur before (admission, June 2017 and February 2018) and after treatment (June 2018 and June 2019) (**c**). Neutralization of pre-adapted phage M1 by antibodies produced by the patient's adaptive immune system upon phage treatment. Bars represent the mean values of biological replicates ($n = 6$), each represented by a circle, with error bars representing standard deviations of the means (**d**). Source data are provided as a Source Data file.

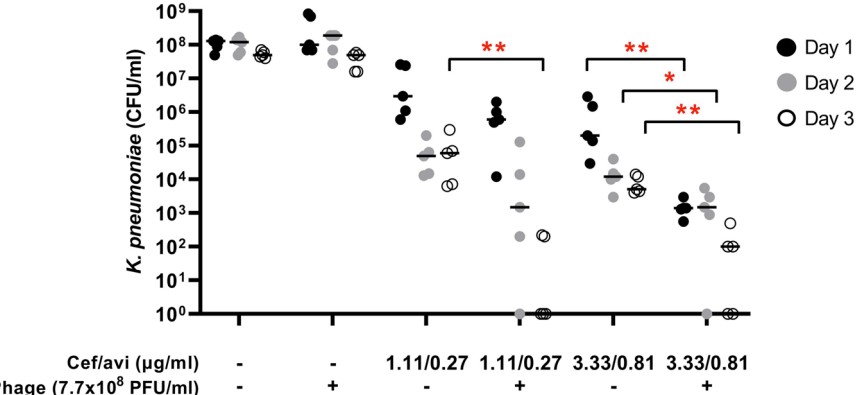

**Fig. 8 Activity of preadapted phage M1, ceftazidime/avibactam, and combinations thereof against PDR *K. pneumoniae* isolate Kp36336 residing in mature biofilms.** Seven-day mature biofilms of *K. pneumoniae* isolate Kp36336 were exposed daily for 3 subsequent days to phages (7.7 × 108 PFU), moderate concentrations of ceftazidime/avibactam (just below and just above the MIC value of 2.0/0.5 μg/ml), and combinations thereof. After exposure for 1–3 days, bacteria in the biofilms were recovered by sonication (10 min at 40 kHz) in saline (or saline supplemented with 10 μM ammonium iron (II) sulphate hexahydrate to neutralize residual phage activity) and subsequently plated on Mueller Hinton agar for microbiological determination of bacterial counts. Results are presented as the median (horizontal bars) of biological replicates ($n = 5$), each dot representing the sum of bacterial counts from pools of 6 wells (technical replicates). The significance of the difference in bacterial counts between samples from biofilms exposed to antibiotic/phage combination and samples from biofilms exposed to the antibiotic alone was determined using a two-sided Mann Whitney U tests, *$p < 0.05$ ($p = 0.0238$), **$p < 0.01$ ($p = 0.0079$). Source data are provided as a Source Data file.

prosthetic knee *K. pneumoniae* infection[21]. Whether phages and antibiotics should be used sequentially or simultaneously is still an unresolved question. "Trojan horse" strategies could be considered where phages are used first to destroy biofilms and activate quiescent cells that will become more susceptible to the antibiotics introduced in a second phase. In addition, some phages have the ability to revert resistance against antibiotics[22]. We cannot exclude that in the present case other factors such as the surgical procedure, including the use of rifampicin-IABG, may have contributed to the recovery of the patient. But then again, rifampicin had no in vitro activity against the infecting *K. pneumoniae* strains, suggesting that the rifampicin-IABG alone did not play a major role in the improvement of the patient's *K. pneumoniae* FRI.

Most phage preparations that are currently researched and developed by western companies are defined phage cocktails, which seem to underappreciate a number of phage peculiarities such as target specificity and antagonistic coevolution[23]. In recent randomized controlled trials, these static phage cocktails showed disappointing results, which contrast with those of an increasing number of case studies using phages as adjunctive therapy[24], or preadapted (or even engineered) phages that are more effective against the infecting bacteria[25,26]. The present case study can open a new way of thinking about phage therapy: the use of individually adjusted phage-antibiotic combinations.

## Methods

**Ethical approval**. The research protocol (P2016/516) was approved by the ethical committee of the Erasme Hospital, and the participant gave written informed consent, according to CARE guidelines and in compliance with the Declaration of Helsinki principles. Consent to publish clinical information potentially identifying individuals was also obtained.

**Antibiotic susceptibility**. Antibiotic susceptibilities of *K. pneumoniae* isolates were ascertained using the VITEK 2 system (bioMérieux). Moxifloxacin and tigecycline susceptibilities were determined using the Kirby-Bauer disk-diffusion method (NeoSensitabs, International Medical Products). Categorization (therapeutic interpretation) of minimum inhibitory concentrations (MICs) or inhibition zone diameters were based on European Committee on Antimicrobial Susceptibility Testing (EUCAST) guidelines.

**Genome sequencing and analysis of *K. pneumoniae* isolates**. The genomes of the 2 day-170 isolates (Kp040741 and Kp040762) and of the day-702 *K.*

*pneumoniae* isolate (Kp36336) were sequenced (day-4 isolate Kp1 was not stored), as previously described[27]. Total genomic DNA (gDNA) was extracted from the isolates with the DNeasy UltraClean Microbial kit (Qiagen) according to the manufacturer's instructions. The gDNA was subsequently prepared for Illumina sequencing using the Nextera Flex (Illumina) and sequenced on an Illumina MiniSeq machine using a paired-end approach (2*150 bp). In addition, the gDNA was also prepared for long-read sequencing using the Rapid barcoding kit SQK-RBK004 (Oxford Nanopore Technology) and sequenced on a MinION equipped with a R9.4.1 flowcell (Oxford Nanopore Technology), with Guppy (v3.1.5) as basecaller. The quality of the Illumina sequencing data was assessed using FastQC v0.11.9 and trimmomatic v0.38 for adapter clipping, quality trimming (LEAD-ING:3 TRAILING:3 SLIDINGWINDOW:4:15), and minimum length exclusion (>50 bp). The quality of the Nanopore reads was assessed using Nanoplot v1.28.2, and Porechop v0.2.3 was used for barcode clipping and NanoFilt v2.6.0 for filtering on quality (Q > 8) and length (>500 bp). The genomes were reconstructed using the hybrid assembler Unicycler v0.4.8. Single Nucleotide polymorphism (SNP) calling was done using snippy v4.4.5 against the PGAP annotation of isolate Kp36336. The assemblies were visually inspected using Bandage[28] and the plasmid elements were checked using BLASTn against the NCBI plasmid database (accessed in October 2020). Further functional annotation, antibiotics resistance genes, prophage elements, and genomic islands were annotated using respectively eggNOG-mapper v2.0.0[29], ABRicate v1 (https://github.com/tseemann/abricate, accessed in October 2020), PHASTER[30] (https://phaster.ca, accessed in October 2020), and Island-Viewer 4[31]. Unless specified otherwise, programs were run using the default parameters. The sequence type and capsule type of the strain were determined using the software mlst to query the pubMLST database[32] (https://github.com/tseemann/mlst, accessed in October 2020) and Kaptive v0.7.3[33], respectively. The genomic datasets were deposited in the NCBI databases. The initial Illumina and Nanopore sets of reads are available in the SRA database via the accession numbers mentioned in Supplementary Data 5.

**Phage selection, preadaptation, and propagation**. The lytic activity of 12 *Klebsiella* phage clones from the Eliava IBMV collection against the patient's day-170 isolates Kp040741 and Kp040762 was determined by spot test and the double agar overlay method[34]. To enhance the lytic activity of phages on the patient's isolates, a phage adaptation procedure based on Appelmans' method[10] was applied for 15 rounds, using both day-170 *K. pneumoniae* isolates. The selected and adapted phage was propagated on isolate Kp040762 using the double agar overlay method.

**Phage characterization**. Relevant parameters of phage adsorption and reproduction cycle, such as adsorption time, latent period (LT), and burst size (BS), were determined for the *K. pneumoniae* propagation strain 1a, of clinical origin, using the methods described by Adams[34]. The morphology and size of the phage particles were evaluated using a JEOL 100SX electron microscope (Jeol). Phages were transferred onto carbon-coated copper grids for 30 s, to let the particles settle, and stained with 1% of uranyl acetate for 40 min. The grids were examined at 250,000x magnification. The infectivity and stability of the selected phage in different environments (acidity and temperature) were evaluated on the *K. pneumoniae*

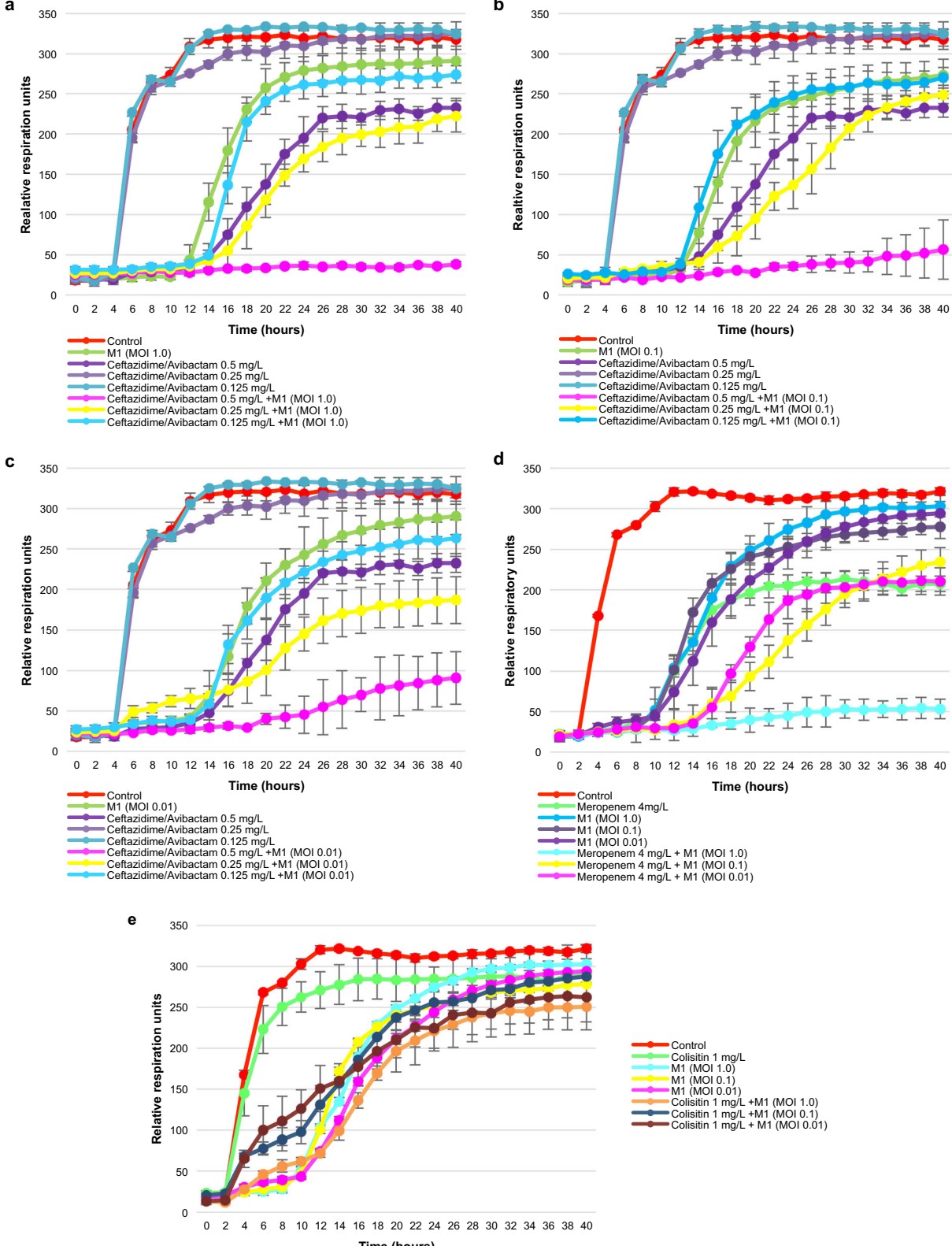

**Fig. 9 Phage-antibiotic synergy.** The activity of preadapted phage M1 at different multiplicities of infection (MOIs), of antibiotics (ceftazidime/avibactam, meropenem, and colistin) at different concentrations, and of combinations thereof, against pandrug-resistant (PDR) *Klebsiella pneumoniae* isolate Kp36336 (isolated 702 days postinjury) were determined. Ceftazidime/avibactam at different concentrations (0.125, 0.25, and 0.5 mg/l) and phage M1 at MOI 1.0 (**a**), 0.1 (**b**), or 0.01 (**c**), and combinations thereof. Meropenem (4.0 mg/l) at MOI 1.0, 0.1, and 0.01 (**d**). Colistin (1.0 mg/l) at MOI 1.0, 0.1, and 0.01 (**e**). Controls consisted of the *K. pneumoniae* isolate Kp36336 without phage M1 or antibiotics. Bacterial proliferation (cellular respiration) is presented. Efficacious phages, antibiotics, and combinations thereof, suppress bacterial proliferation. Results are presented as mean values of biological replicates ($n = 3$) with error bars representing the standard deviations of the means. Source data are provided as a Source Data file.

strain 1a. To determine the impact of acidity, phages were incubated at a titer of $2 \times 10^7$ PFU/ml in lysogeny broth (LB) with pH ranging from 2.0 to 14.0 at 37 °C for 1 h. The proportion of persistent phage particles showing infectivity towards the host strain was determined by the double agar overlay method[34]. To determine the impact of temperature, phages were incubated at a titer of $3 \times 10^7$ PFU/ml in LB at temperatures ranging from 20 to 75 °C (with 5 °C steps) for 1 h. After incubation, the number of viable phage particles was determined by the double agar overlay method on the host strain[34].

**Bacterial growth kinetics in the presence of phage M1 and/or antibiotics.**
Bacterial respiration was measured using the Omnilog system (Biolog, Hayward, CA, USA)[25,35]. The growth kinetics of *K. pneumoniae* isolates Kp040741, Kp040762, and Kp36336 were assessed in the presence of phage M1 (at different MOIs), relevant antibiotics (ceftazidime/avibactam, meropenem, and colistin, at different concentrations), and phage-antibiotic combinations. Experiments were done in 96-well plates (Thermo Fisher Scientific, Roskilde, Denmark) in a final volume of 200 µl of LB supplemented with 100-fold diluted tetrazolium dye mix A, according to the manufacturer's instructions. Bacterial cells were added at a concentration of $10^4$ CFU/well, calculated based on optical density (OD, at 600 nm) measurements (with an OD of 0.5 corresponding to $4 \times 10^8$ CFU/ml, on average), which were validated using a classical plate culture method. Antibiotics and the phage M1 were diluted and used according to the concentrations and MOIs indicated in the captions of Figs. 6 and 9). The titer of the phage M1 was also confirmed after each experiment using the classical double agar overlay method[34]. Plates were incubated at 37 °C for 72 h and reduction (causing a color change) of the tetrazolium dye due to bacterial respiration (during growth) was recorded every 15 min by the Omnilog system. Experiments were performed in triplicate (biological replicates).

**Phage genome sequencing and analysis.** Phage DNA was extracted from a high-titer stock of the originally selected variant and the pre-adapted variant of phage M1 and subsequently sequenced using in-house MiniSeq Illumina sequencing as described in Makalatia et al.[36]. Genome assembly was performed using SPAdes (Galaxy v3.12.0 + Galaxy1)[37]. Next, using MEGA X v10.1.2, the phage genome was aligned to *Klebsiella* phage vB_KpnM_KP15, which was identified as the closest type species by BLASTn against the NCBI nucleotide database (accessed in March 2021). Finally, the phage genome was annotated using RASTtk (PATRIC v3.6.12)[38] and manually curated by BLASTp against the NCBI protein database (accessed in March 2020). The genome was screened for tRNAs using tRNAscan-SE 2.0 and was visualised as a genome map using EasyFig 2.2.2[39]. We used VirulenceFinder v2.0[40], ABRicate v1 (https://github.com/tseemann/abricate, accessed in October 2020) and manually verified the annotated proteins to screen for lysogeny-related proteins. The genomes of the originally selected variant and the pre-adapted variant of phage M1 used in therapy were analyzed for distinguishing mutations using the variant caller snippy v4.6.0. The mutated protein was modelled using Alpha-Fold v2.0[41] (https://doi.org/10.1038/s41586-021-03819-2, accessed in September 2021).

**Biofilm assay and exposure to ceftazidime/avibactam, phages and combinations thereof.** Frozen stocks of PDR *K. pneumoniae* strain Kp36336 in 20% (v/v) glycerol (Thermo Fisher Diagnostic BV) were thawn and then spread on plates with trypticase soy agar with 5% (v/v) sheep blood (bioMérieux) and cultured overnight at 37 °C. Subsequently, several colonies were resuspended in tryptic soy broth (Oxoid Ltd) and cultured for 2.5 h at 37 °C under slow rotation (200 rpm). The mid-logarithmic phase bacteria were harvested by centrifugation (1,000 x *g* for 10 min), washed with phosphate-buffered saline (PBS; pH 7.4), and then diluted (based on the optical density at 600 nm) in brain heart infusion broth (Oxoid Ltd) to a concentration of $1 \times 10^7$ CFU/ml. Next, 100 µl of the bacterial suspension were transferred to 96-wells polystyrene wells flat-bottom plates (Greiner Bio-One) which were then sealed with breathable Ryon film sealers (VWR European). The bacteria were cultured for 7 days at 37 °C in a plastic box to maintain a humidified environment and medium controls were included to monitor possible contamination. Thereafter, the wells were washed twice with saline to remove planktonic bacteria and the biofilms were exposed daily for 3 days to fresh preparations of ceftazidime/avibactam (Pfizer, range: 0–90/22.5 µg/ml), phage (range: $7.7 \times 10^7$ to $7.7 \times 10^{10}$ PFU/ml), or combinations thereof. After 1, 2 and 3 days, the biofilms in 96-wells plates were washed with saline and then sonicated (10 min at 40 kHz) in 100 µl of saline for 10 min. Finally, serial dilutions of the bacterial suspensions were plated on Mueller Hinton agar plates (Oxoid Ltd) for CFU enumeration. In order to neutralize residual bacteriolytic activity in the samples from biofilms exposed to phages bacterial suspensions were diluted in saline supplemented with 10 µM ammonium iron (II) sulphate hexahydrate (Sigma-Aldrich) prior to the assessment of the bacterial counts as described above.

**Phage administration.** At the end of the surgical intervention, 100 ml of the phage preparation ($10^8$ PFU/ml) diluted in 1,000 ml of sterile normal saline (0.9% NaCl) were instilled locally in the surgical wound via a catheter. The catheter was left in place to facilitate local applications of 20 ml of undiluted phages, three times

per day for 5 days (Fig. 1). This regimen was based on previous experiences at the Eliava Institute in Tbilisi (Georgia).

**Phage neutralization by patient sera.** The ability of the patient's serum to neutralize the phages was evaluated according to Adams[34] with some modifications. Whole blood samples were collected 0, 1, 2, 5, 8, 18, 161 days, and 3 years post-phage application. Blood was allowed to clot for a minimum of 30 min in a vertical position and then centrifuged at room temperature in a swinging bucket rotor for 10 min at 2,000 x *g*. Serum samples were stored at −80 °C ± 5 °C. To assess the effect of the sequential serum samples on phage lytic activity, 0.9 ml of the diluted sera (1:100) were mixed with 0.1 ml of the phage at a concentration of $2 \times 10^7$ PFU/ml and incubated at 37 °C for 30 min. After incubation, the phages were titered with 2 of the patient's *K. pneumoniae* strains (Kp040741 and Kp040762) to determine the remaining number of non-neutralized phages and thus also the proportion of neutralized phages. Each sample was tested in triplicate on each bacterial strain and the mean value and standard deviation were determined for each serum sample.

**Reporting summary.** Further information on research design is available in the Nature Research Reporting Summary linked to this article.

## Data availability

The Illumina and Nanopore sequencing datasets of the *Klebsiella pneumoniae* genomes were deposited in the Sequence Read Archive (SRA) database under accession codes SRR13350373-SRR13350378. The genome of isolate Kp36336 was deposited in the GenBank database under accession codes CP066511-CP066515. The Illumina data of the wild type and the adapted phages can be accessed via the NCBI SRA database under the accession codes SRR15604575 and SRR15694574, respectively. The genome sequence of phage vB_KpnM_M1 was deposited in the GenBank database under accession code MW448170. Source data are provided with this paper. The authors declare that all other data supporting the findings of this study are available within the article and its Supplementary Data files. Source data are provided with this paper.

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

## Acknowledgements

We thank A. Vanderkelen for helping facilitate the application of phages. We thank S. Djebara for sample collection. We thank J. Onsea for her critical reading of the manuscript.

## Author contributions

A.E. and M.M. devised the phage therapy protocol. A.E., M.J., and M.H. contributed to patient data collection, analysis, and interpretation. N.B., L.L., L.A., and L.K. contributed to phage selection, characterization, pre-adaptation and production. C.L., V.V.N., and J. Wagemans contributed to bacterial and phage genome sequencing and analysis. M.M. contributed to phage neutralization assays and analysis. J. Wubbolts and P.N. contributed to biofilm assays and data analysis. A.E., M.H., N.C., M.K., R.L., M.D.B., and J.P.P. contributed to literature search, study design, data interpretation, and writing. J.P.P. prepared the original draft.

## Competing interests

The authors declare no competing interests.
