## [Peer Review File · Nature Communications]

Combination of pre-adapted bacteriophage therapy and antibiotics for treatment of fracture-related infection due to pandrug-resistant *Klebsiella pneumoniae*Reviewers' Comments:

Reviewer #1:

Remarks to the Author:

This is a well-written case report of a young patient with a complex course of a femoral traumatic non-union with a polymicrobial infection including multidrug resistant *Klebsiella* involving hardware in the setting of soft tissue reconstruction.

Of the many strengths of this paper is the richly detailed clinical course, particularly the numerous and severe adverse events suffered by the patient from conventional antibiotics, in addition to the prolonged and multiple relapses and failures with conventional medical and surgical treatments.

The authors make a successful case for the clinical, radiographical, and microbiological improvement with objective data. They address the difficulties of proving benefit of a therapy that is often used adjunctively, in combination with other treatment modalities, such as the addition of ceftazidime/avibactam and the use of rifampicin-IABG. Multiple treatment modalities are often combined due to desperation in end-stage cases such as this. I believe the phage therapy field as a whole would benefit from this publication.

Please provide the following details/clarification:

-Line 99: Can the authors briefly mention why the mucor infection resulted in gastrectomy and splenectomy?

-Lines 142-144: Please provide details on how the phages were applied locally for 6 days. Was there a catheter left in place or was there a direct injection into the site? Or another method?

-Line 144: How was the mode of administration selected? Was there any consideration of combination of systemic (intravenous) phage plus local administration?

-Lines 177-179: What is the authors' hypothesis as to why the patient improved despite the development of neutralizing antibodies?

-Line 195: Please briefly describe what is meant by "regained autonomy". Is the patient currently ambulatory?

-Lines 239-240: The authors discuss the ambiguities of the importance of phage/antibiotic sequence of administration. Please describe the sequence of the phage and antibiotic administration in this case and the rationale.

Gina Suh, MD

Reviewer #2:

Remarks to the Author:

Eskenazi et al. describes here a highly likely successful treatment using phage therapy of a fracture-related infection of highly resistant *K. pneumoniae*. The authors performed genome analysis of both phage and bacterial isolates and monitored the patient for 3 years.

In my opinion, this is an impressive example of the remarkable abilities of phages to treat, otherwise untreatable, infections. Such cases, raise the hope that the current trend which is expected by 2050 to lead antibiotic resistant bacteria to be the 1st medical-related cause of death, will be change.

I also agree with the authors view that one of the key of success in phage therapy is a careful and accurate pre-matching of the phages and antibiotic to the target bacteria.

Major Comment

None

Minor comments and questions

- How was it test that "its genome does not encode proteins associated with toxicity, lysogeny or antibiotic resistance"?
- No line numbers are added to the manuscript.
- A space is missing in: "Phage therapy, the use of (bacterio)phages"
- In section "Genome sequencing and analysis of K. pneumoniae isolates", add the major parameters of the software used (or mentioned "with default parameters"). In addition, add the reference or web sites to each software used. Version is missing in: eggNOG-mapper, abricate, PHASTER.
- About the "training" which is a major issue here and appears in the title, can the authors elaborate more on the genotypic and phenotypic changes happened to the "trained" phage compared to the original phage from Georgia? Also, did the parameters such as adsorption curve, single-step growth curve and pH stability, presented in Extended Data Fig. 1 were determined for the trained phage, or for the original variant? Is the presented phage genome sequencing sent to GeneBank is that of the original or to the trained phage variant? Can the authors supply additional analyses of the changes in phage genotypic or phenotypic characteristics achieved by the training?
- The authors state that the pre-adapted phage was equally active against the day-702 bacterial isolate. How was this evaluated? Please provide more details. The growth kinetics plot described in the text better be presented as a figure.
- Was the phage-antibiotic combination tested in-vitro prior to treatment initiation, as can be concluded from the article's title?
- From figure 1 it seems that it took almost a year and a half from EC approval to the introduction of phage treatment. Would that reflect that there was no active infection between the 2 periods? Could it be that the isolates in Q1 2018 are not related to the initial infection? can the author clarify that?
- I recommend that the isolation dates and the full antimicrobial susceptibility (with MIC) will be added in the Supp excel table of all KP isolates.
- Obviously with such infection there is both planktonic and biofilm formation. However, only biofilm kinetic is presented. Can the author add the growth kinetic for planktonic growth? This will help understanding the mechanism for which this treatment that included both phage and antibiotics resulted in clinical improvement of the patient.
- Several antibiotics were used in the matching. However, only ceftazidime/avibactam are presented. Can the authors present all combination of antibiotics and phages?
- Does the data presented in figure 3 was done to 3 or 4 KP isolates presented in EX table 1?
- Can the authors specify the anticipated concentration or PK/PD of ceftazidime/avibactam in the regimen used in this patient and its correlation to concentration presented in figure 3?
- What could be the reason that the inflammatory parameters were "normal" prior to surgical procedure (table 2a)?
- The "biological parameters" in figure 2a do not provide sufficient details and do not allow any conclusion drawing regarding the treatment success (only two measurements of WBC and CRP with no clear trend, only one measurement of ESR). This section should provide much broader details for these parameters across different timepoints. Moreover, in order to suggest non inflammation or non-infection processes at the site of infection an MRI or preferably PET CT are needed. I suggest that if these modalities were performed, they should be added here. If this data is not available, in my opinion, the interpretation of "non inflammation or non- infection" at the site of infection is not valid. The presented result of June 2019 of callus formation with partial non-union or double bone, the inflammation markers prior to surgical procedure and the ability to walk on this leg, may only suggest infection- suppression and the question if chronic osteomyelitis remained is still open. Please elaborate on that in the text.
- Due to the differences in EX data figure 3, I suggest to present also either MLST; capsule typing or core genome (for example with Roary) analysis. Additional data on major fragment differences and rearrangements should be presented using Mauve Artemis or similar programs.
- Some wording in the manuscript are generally not used in medical English and were probably translated. Please revise the English.
- Figure 2 a add the time frame for the "Before surgery" column
- I think that some references should be added including prior published bone infections with KP that were treated with phages. Also the summery of experience from Lyon and IPATH which were

published recently.

- In continuation of the previous comment, perhaps a table which summarize the osteomyelitis-phage therapy cases of polymicrobial and single bacterial may be added for comparison.
- The authors mentioned that Phage M1 showed a broad host range (~65%) against clinical isolates from various origins. It would be interesting to mention the distribution of the isolates according to their geographical origin.
- In the biofilm model, was fresh media inserted to the wells after planktonic bacterial removal and before treatment administration? The authors should describe the protocol more clearly.

Ronen Hazan

Reviewer #3:

Remarks to the Author:

This article suggests that combining phage therapy with antibiotics may be a more efficient means of eradicating multidrug resistant bacterial infections than either treatment alone. Therefore, this could be an important contribution to the ongoing discussion on phage-antibiotic synergy.

The authors cite an article by Grimaldi et al published in Lancet Infect Dis in 2017 describing a successful treatment of infection (both bacterial and fungal) with immunotherapy. It appears that the same patient has now been treated with phage and antibiotics. "No residual infection" was noted in the patient who was discharged earlier from the clinical ward but now the authors describe "therapeutic failure" and "therapeutic dead end". This requires some explanation and comments.

Moreover, if the patient showed signs of improvement within two days of phage therapy why a different antibiotic regimen has been applied?

A.Gorski

POINT-BY-POINT RESPONSE TO THE REVIEWERS' COMMENTS

First of all, we would like to thank the reviewers for their time and constructive comments. We addressed their concerns, using Track Changes, in a revised manuscript. We feel that addressing the reviewers' comments and questions improved the quality of our manuscript.

Please find below our point-by-point answers to the reviewers' comments and questions.

Reviewer #1:

- Line 99: Can the authors briefly mention why the mucor infection resulted in gastrectomy and splenectomy?

Due to extended necrosis of the stomach and the spleen, gastrectomy and splenectomy was performed conform the following recommendation: Cornely OA, et al., Mucormycosis ECMM MSG Global Guideline Writing Group. Global guideline for the diagnosis and management of mucormycosis: an initiative of the European Confederation of Medical Mycology in cooperation with the Mycoses Study Group Education and Research Consortium. *Lancet Infect Dis.* 2019 Dec;19(12):e405-e421). This information was added to the revised manuscript.

- Lines 142-144: Please provide details on how the phages were applied locally for 6 days. Was there a catheter left in place or was there a direct injection into the site? Or another method?

Phages were indeed applied through a catheter left in place. We added this information to the revised manuscript.

- Line 144: How was the mode of administration selected? Was there any consideration of combination of systemic (intravenous) phage plus local administration?

It was indeed considered, but the medical team only reached a consensus for local use. We added this information to the revised manuscript.

- Lines 177-179: What is the authors' hypothesis as to why the patient improved despite the development of neutralizing antibodies?

We hypothesize that phages played a role in the biofilm clearance and had an antibacterial effect in synergy with antibiotics and that the neutralizing antibodies didn't hamper that process because they appeared after the treatment was completed (see Extended Data Fig. 7).

- Line 195: Please briefly describe what is meant by "regained autonomy". Is the patient currently ambulatory?

The patient is ambulatory, walks with crutches and was even able to participate in sports events, for instance a stage of the Tour de France (<https://www.france24.com/fr/sports/20210706-tour-de-france-11-victimes-du-terrorisme-%C3%A0-l-assaut-du-mont-ventoux>). We added more information on the patient's current status in the revised manuscript.

- Lines 239-240: The authors discuss the ambiguities of the importance of phage/antibiotic sequence of administration. Please describe the sequence of the phage and antibiotic administration in this case and the rationale.

Following the study design, the patient was already under a long-term course of antibiotics that showed limited *in vitro* activity against the pandrug-resistant (PDR) *Klebsiella pneumoniae* strain. Since the recommendation for treatment of fracture-related infection (FRI) prescribe the use of antibiotics after surgery, it was decided to continue this antibiotic treatment. Phage therapy was added as an additional anti-bacterial treatment. When later, shortly after completion of phage therapy, susceptibility testing showed that ceftazidime-avibactam was effective against the PDR *K. pneumoniae* strain, it was decided to use this newly available (under certain conditions) antibiotic in order to maximize potential phage-antibiotic synergy. So, in conclusion, phages were used

simultaneously with some conventional antibiotics and prior to the newly available ceftazidime-avibactam treatment. We added this information to the revised manuscript.

Reviewer #2:

- How was it tested that “its genome does not encode proteins associated with toxicity, lysogeny or antibiotic resistance”?

We added the following statement to the supplementary materials section: “We used VirulenceFinder v2.0, ABRicate v1, and manually verified the annotated proteins to screen for lysogeny-related proteins.”

- No line numbers are added to the manuscript.

We added line numbers.

- A space is missing in: “Phage therapy, the use of (bacterio)phages”.

We could not find the missing space in the above-mentioned sentence.

- In section “Genome sequencing and analysis of *K. pneumoniae* isolates”, add the major parameters of the software used (or mentioned “with default parameters”). In addition, add the reference or web sites to each software used. Version is missing in: eggNOG-mapper, abricate, PHASTER.

All software was run with default parameters (usually, only deviations from default parameters are mentioned). We added this information. We also added references for Bandage, PHASTER, eggNOG-mapper, and IslandViewer 4. We added versions of eggNOG-mapper and ABRicate. Please note that PHASTER is a web application (<https://phaster.ca>) which does not have a software version number. The web application was accessed in October 2020 and the URL was added.

- About the “training” which is a major issue here and appears in the title, can the authors elaborate more on the genotypic and phenotypic changes happened to the “trained” phage compared to the original phage from Georgia? Also, did the parameters such as adsorption curve, single-step growth curve and pH stability, presented in Extended Data Fig. 1 were determined for the trained phage, or for the original variant? Is the presented phage genome sequencing sent to GeneBank is that of the original or to the trained phage variant? Can the authors supply additional analyses of the changes in phage genotypic or phenotypic characteristics achieved by the training?

We compared the genomes of the originally selected variant and the pre-adapted phage variant of M1 that was used in therapy. A missense mutation (Thr281Arg) in the loop region of the hinge connector of the distal tail fiber protein (Extended Data Fig. 2 and 3 and Supplementary Table 1), which may cause alterations in the phage receptor, is suspected to be at the basis of the observed improved lytic activity against the patient's day-170 *Klebsiella pneumoniae* isolates. However, conclusive proof of this genotype-phenotype association would require engineering the originally selected phage M1 isolate to reintroduce this mutation independently, which we have not been able to achieve to date.

Parameters such as adsorption curve, single-step growth curve and pH stability were determined for the adapted phage on the propagation strain A1, and not the patient's strain, as emphasized in the revised manuscript.

- The authors state that the pre-adapted phage was equally active against the day-702 bacterial isolate. How was this evaluated? Please provide more details. The growth kinetics plot described in the text better be presented as a figure.

This was evaluated by defining efficiency of plating (EOP) using the spot test and the double agar overlay method. In the current version of the manuscript, we also added the growth kinetics of

three isolates (incl. the day-702 isolate) in the presence of phage M1 (at different MOIs), as evaluated using the OmniLog imaging system (see the newly added Extended Data Fig. 6).

- Was the phage-antibiotic combination tested in-vitro prior to treatment initiation, as can be concluded from the article's title?

It was our intention to suggest that pre-adapted phage and antibiotics were used in combination, not that their combination was tested or adapted prior to therapy. The *in vitro* activity of the phages and antibiotics were determined prior to treatment, but separately, not in combination. The *in vitro* phage-antibiotic synergy was documented after completion of the therapy. To avoid misunderstandings, we adapted the title of the revised manuscript.

- From figure 1 it seems that it took almost a year and a half from EC approval to the introduction of phage treatment. Would that reflect that there was no active infection between the 2 periods? Could it be that the isolates in Q1 2018 are not related to the initial infection? can the author clarify that?

The reason for the long interval between EC approval and phage therapy is mentioned in the manuscript: "However, due to lack of consensus among the treating physicians, phage therapy was put on hold." The reason was not infection-related. There was still an active infection.

- I recommend that the isolation dates and the full antimicrobial susceptibility (with MIC) will be added in the Supp excel table of all KP isolates.

Isolation dates and MICs were added to the Extended Data Table 1.

- Obviously with such infection there is both planktonic and biofilm formation. However, only biofilm kinetic is presented. Can the author add the growth kinetic for planktonic growth? This will help understanding the mechanism for which this treatment that included both phage and antibiotics resulted in clinical improvement of the patient.

Planktonic phage-antibiotic synergy experiments were performed using the OmniLog imaging system and the results were added (the newly added Extended Data Fig. 8) and discussed in the revised manuscript.

- Several antibiotics were used in the matching. However, only ceftazidime/avibactam are presented. Can the authors present all combination of antibiotics and phages?

Possible synergy between phage M1 and the other relevant antibiotics that were used in combination (meropenem and colistin) was analyzed using the OmniLog imaging system. The results were added (the newly added Extended Data Fig. 8) and discussed in the revised manuscript.

- Does the data presented in figure 3 was done to 3 or 4 KP isolates presented in EX table 1?

As stated in the "methods" section and in the caption of Figure 3, the data presented relates to PDR *K. pneumoniae* isolate Kp36336, the day-702 isolate that was present at the time of phage therapy. Day-4 and day-170 isolates were considered less relevant as they were isolated more than one year prior to phage therapy.

- Can the authors specify the anticipated concentration or PK/PD of ceftazidime/avibactam in the regimen used in this patient and its correlation to concentration presented in figure 3?

The ceftazidime-avibactam regimen that was used in the patient (2 g/0.5 g, q8) was conform the recommendations of the manufacturer. According to the manufacturer, peak plasma concentration are 90.4 µg/ml and 14.6 µg/ml for ceftazidime and avibactam, respectively (according to the FDA-approved labeling information). The ceftazidime-avibactam concentrations that are presented in Figure 3 (1.1/0.27 and 3.3/0.81 µg/ml) are just below and just above the ceftazidime/avibactam MIC value of 2.0/0.5 µg/ml, obtained for the PDR Kp36336 isolate. This information was added to the revised manuscript.

- What could be the reason that the inflammatory parameters were “normal” prior to surgical procedure (table 2a)?

As shown in the meta-analysis (van den Kieboom et al. Diagnostic accuracy of serum inflammatory markers in late fracture-related infection: a systematic review and meta-analysis. *Bone Joint J.* 2018 Dec;100-B(12):1542-1550), CRP, leucocyte rate and erythrocyte sedimentation rate are, unfortunately, not sufficiently accurate markers of fracture related infection (FRI).

- The “biological parameters” in figure 2a do not provide sufficient details and do not allow any conclusion drawing regarding the treatment success (only two measurements of WBC and CRP with no clear trend, only one measurement of ESR). This section should provide much broader details for these parameters across different timepoints. Moreover, in order to suggest non inflammation or non- infection processes at the site of infection an MRI or preferably PET CT are needed. I suggest that if these modalities were performed, they should be added here. If this data is not available, in my opinion, the interpretation of “non inflammation or non- infection” at the site of infection is not valid. The presented result of June 2019 of callus formation with partial non-union or double bone, the inflammation markers prior to surgical procedure and the ability to walk on this leg, may only suggest infection- suppression and the question if chronic osteomyelitis remained is still open. Please elaborate on that in the text.

Thank you for pointing out this very complicated aspect of chronic fracture-related infection (FRI). While biological inflammatory markers and radiological signs are considered as suggestive criteria of FRI, the confirmatory criteria are fistula - sinus and wound breakdown and/or the presence of pus, referring to the consensus definition of FRI published in 2018 (Metsemakers WJ et al. Fracture-related infection: A consensus on definition from an international expert group. *Injury.* 2018 Mar;49(3):505-510). This was emphasized in the revised manuscript.

- Due to the differences in EX data figure 3, I suggest to present also either MLST; capsule typing or core genome (for example with Roary) analysis. Additional data on major fragment differences and rearrangements should be presented using Mauve Artemis or similar programs.

The sequence type and capsule type of the strain were determined using the software mlst (to query the pubMLST database) and Kaptive v0.7.3, respectively. The mlst analysis revealed that our strain belonged to sequence type ST839, which allowed us to refer to an ST839 outbreak in Iran, but no whole genome sequencing data is (publicly) available for that cluster. Additional information was added to the “methods” section.

- Some wording in the manuscript are generally not used in medical English and were probably translated. Please revise the English.

Maya Hites, one of the authors of this manuscript, and apparently also reviewer 1, both native English speaking Medical Doctors, feel that the used wording is correct.

- Figure 2 a add the time frame for the "Before surgery" column.

A time window was added.

- I think that some references should be added including prior published bone infections with KP that were treated with phages. Also the summery of experience from Lyon and IPATH which were published recently.

We added a recent relevant reference (Cano EJ et al. Phage Therapy for Limb-threatening Prosthetic Knee *Klebsiella pneumoniae* Infection: Case Report and *In Vitro* Characterization of Anti-biofilm Activity. *Clin Infect Dis.* 2021 Jul 1;73(1):e144-e151). It is not our intention to review this type of phage application (e.g., adding an exhaustive list of cases), since this is a case report (brief communication).

- In continuation of the previous comment, perhaps a table which summarize the osteomyelitis-phage therapy cases of polymicrobial and single bacterial may be added for comparison.

See the previous answer.

- The authors mentioned that Phage M1 showed a broad host range (~65%) against clinical isolates from various origins. It would be interesting to mention the distribution of the isolates according to their geographical origin.

This information is provided in the caption of Extended Data Fig. 1.

- In the biofilm model, was fresh media inserted to the wells after planktonic bacterial removal and before treatment administration? The authors should describe the protocol more clearly.

Yes, the mature biofilms were washed twice with sterile saline to remove planktonic bacteria. Thereafter, phages, antibiotic, or combinations thereof, were applied on top of the biofilms. The relevant paragraph in the “methods” section was adapted to clarify this.

Reviewer #3:

- The authors cite an article by Grimaldi et al published in Lancet Infect Dis in 2017 describing a successful treatment of infection (both bacterial and fungal) with immunotherapy. It appears that the same patient has now been treated with phage and antibiotics. “No residual infection” was noted in the patient who was discharged earlier from the clinical ward but now the authors describe “therapeutic failure” and “therapeutic dead end”. This requires some explanation and comments.

In the article by Grimaldi the “no residual infection” referred to the mucormycosis (the subject of the paper). A persistent sinus track discharge of pus and no consolidation of femoral fracture after months of antibiotic treatment, motivated our “therapeutic dead-end” conclusion.

- Moreover, if the patient showed signs of improvement within two days of phage therapy why a different antibiotic regimen has been applied?

The protocol approved by the medical team (consensus) provided for the application of a combination of phage with the “standard” FRI treatment procedure, which consist of a long-term course of antibiotics. The change of antibiotics was decided when, shortly after phage therapy, susceptibility testing showed that a new available antibiotic (ceftazidime/avibactam) showed *in vitro* activity. The medical team felt that the patient could not be denied an application of an antibiotic that showed *in vitro* activity, even though phage therapy had resulted in signs of improvement.

Reviewers' Comments:

Reviewer #2:

Remarks to the Author:

The authors answered all my questions and revised the manuscript accordingly. I don't have more comments